# Health-Related Quality of Life and Bereavement in the 2019 Georgia Behavioral Risk Factor Surveillance System

**DOI:** 10.3390/bs14121213

**Published:** 2024-12-18

**Authors:** Changle Li, Toni P. Miles

**Affiliations:** 1School of Health Management, Fujian Medical University, Fuzhou 350122, China; changle.li@uga.edu; 2College of Public Health, University of Georgia, Athens, GA 30602, USA; 3Rosalynn Carter Institute for Caregivers, Americus, GA 31709, USA

**Keywords:** bereavement, Health-Related Quality of Life, BRFSS, smoking, obesity, mediation, mental health, physical health

## Abstract

Poor Self-Rated Health (SRHp) is part of a four-item scale for self-assessment. SRH from the 2019 Behavioral Risk Factor Surveillance Survey (BRFSS) is used to test hypotheses linking population-level well-being influenced by bereavement due to the death of a close friend or relative. By linking the prevalence rates of population-level well-being with exposure to bereavement, we extend our knowledge of this exposure beyond single-person studies. SRHp and bereavement were asked about in the 2019 field survey of 7354 adults aged 18 years and older. Multiple imputation was applied to handle missing values. Data modeling included adjusted logistic regression and mediation analyses. In the total sample, the prevalence rate of SRHp was 23.1% and the rate of bereavement was 45.5%. The SRHp subgroup had a significantly greater prevalence of bereavement (24.9% versus 21.6% compared to all other SRH categories combined). Elevated bereavement rates were also observed for the group with poor physical health (18.3% versus 14.9%) and poor mental health (17.5% versus 12.8%). Multiple losses (three or more deaths) increased the risk for SRHp by 42% in adjusted logistic models. Although these are cross-sectional data, the analyses provide evidence that bereavement is more common among people engaged in negative health behaviors. In mediation analyses, bereavement increases the prevalence of SRHp by 56.8% through an indirect effect on smoking. Bereavement also increased the rate of SRHp by 40.7% through an indirect effect on obesity. More research is needed to evaluate the association between bereavement and health behavior.

## 1. Introduction

The period after the death of a member of one’s social network is the working definition of bereavement [1]. It is a state of being that is distinct from grief—the emotional response to the death [1]. There is evidence that bereavement has both short- and long- term influences on the domains of Health-Related Quality of Life (HRQoL). Health-related quality of life (HRQoL) refers to “how well a person functions in their life and his or her perceived well-being in physical, mental, and social domains of health” [2,3]. Scattered reports suggest higher prevalence rates of poor mental health among bereaved young adults and teens [2]. Bereavement after a sudden cardiac death [4] or cancer death [5] negatively impacts the health of surviving family members and coworkers. While this evidence can be used to design clinical support for individuals and families, it does not translate directly into population vitality. HRQoL is the most common scale for measurement of subjective well-being for both individuals and in population-level surveillance surveys [6]. The short- and long-term health effects of bereavement on population health is indirectly inferred from samples and case studies. The biases associated with case study data can be reduced with direct assessment of bereavement’s influence on population-wide HRQoL prevalence rates using an annual surveillance survey structure that is designed to recruit a representative sample of the population. However, surveillance surveys do not typically ask participants about bereavement.

When HRQoL is applied to individuals, it is a self-assessment. The availability of self- assessed HRQoL in patient registries has led to the development of longer and more detailed clinical tools like Patient-Reported Outcomes Measures (PROMs) [3], Patient-Reported Outcomes Measurement Information System (PROMIS) [7], and International Survey of People Living with Chronic Conditions (OECD-PaRIS) [8] to guide ongoing care of chronic health conditions. The potential for HRQoL to serve as a metric for population-level health surveillance is currently being tested in a clinical trial by the Organization for Economic Cooperation and Development (OECD)/World Bank [9]. In the United States, population-level HRQoL is a routine part of the annual Behavioral Risk Factor Surveillance Survey (BRFSS) conducted by all 50 U.S. states and three territories (Puerto Rico, the District of Columbia, and Guam). The BRFSS reports HRQoL as categorical prevalence rates for each level of response to the four domains. The U.S. Commonwealth of Puerto Rico (PR) published its HRQoL prevalence rates from their 2019 BRFSS but bereavement was not included in the survey [10]. In this representative sample of PR residents, poor HRQoL was prevalent across the four domains. Consider the domain of poor self-rated health (27%). When these rates are applied to the population of 2.4 million adults aged 18 and older, there are 674,253 Puerto Rican adults aged 18 years and older reporting poor self-rated health.

Dynamic events—such as the occurrence of Hurricane Maria (2017)—were not included in the adjusted models in the PR BRFSS report. Typically, the BRFSS does not measure the occurrence of dynamic events such as financial crisis [11], or post-disaster trauma [12]. To test the influence of bereavement in its 2019 BRFSS, the state of Georgia incorporated a three-question module querying respondents about the occurrence of deaths of friends and family during the 24 months prior to survey. These items are the basis for this test that investigates the association between population-level HRQoL and recent bereavement.

Exposure to the loss of friends and family is a risk for subsequent mortality [13] as well as a decline in physical health [14,15]. Furthermore, bereavement is life’s most stressful event and is associated with increased health care utilization [16] and obesity [17]. The combination of an annual BRFSS core with the new module focused on bereavement can overcome the biases of previous studies on bereavement and health [18]. To fully assess population-level health and bereavement, we apply statistical models to examine the association between population-level HRQoL domains and bereavement.

## 2. Materials and Methods

### 2.1. Study Design

The annual BRFSS is a cross-sectional and telephone interview of U.S. residents in all 50 states plus the areas of Puerto Rico, the District of Columbia, and Guam. Georgia has been part of the system since 1984. In each location, respondents were randomly selected from each household’s non-institutionalized adult population aged 18 years and older to generate weighted population estimates. Each is contacted using list-assisted, random digit dialing. Respondents with either landline telephones or cellular phones are eligible to participate.

The BRFSS has a core set of questions asked by all states. States are also encouraged to add topics of specific interest [19]. During the 2019 BRFSS, only the state of Georgia added a special module containing three items to measure the number of people experiencing the deaths of friends, family, or both in the period between 2018 or 2019. The statewide survey typically begins in late December/early January and is completed by May of the following year. After final review and cleaning of data by the Centers for Disease Control and Prevention, the core data are released in August. Special state modules require further evaluation for completeness and is released in September or October. The 2019 Georgia BRFSS was initiated in December 2019 and completed in May 2020. The data from the common core were made available in August 2020. The data from the Georgia Bereavement Core became available in October 2020. An assessment of response rates and standard errors associated with the 2019 bereavement module [18]. The appendix contains a table listing the variables included in the present analyses along with the number of total respondents and the percentage with complete answers to both the core and bereavement modules combined.

### 2.2. Sample

The 2019 Georgia BRFSS includes a total unweighted sample of 7354 people. There were 5206 people (70.79%) responding to the bereavement module (both Yes or No) to the question about experiencing the death of family, friends, or both categories of relationship in 2018 and 2019. Due to the application of imputation methods, this report uses 7354 as the denominator for all analyses.

### 2.3. Measures

#### 2.3.1. Health-Related Quality of Life (HRQoL)

The BRFSS questionnaire asks for an estimate of the number of unhealthy days as indicators of HRQoL. HRQoL is measured with four simple questions about self-rated health, physical health, mental health, and activity limitations. Self-rated health has five categories: ‘Would you say that in general your health is excellent, very good, good, fair, or poor?’ Physical and mental health are measured separately with these questions: ‘Now thinking about your physical health, which includes physical illness and injury, for how many days during the past 30 days was your physical health not good?’ and ‘Now thinking about your mental health, which includes stress, depression, and problems with emotions, for how many days during the past 30 days was your mental health not good?’ Limitation in activity is an estimate: ‘During the past 30 days, for about how many days did poor physical or mental health keep you from doing your usual activities, such as self-care, work, or recreation?’ These four HRQoL questions have been proven to be easy to administer, valid, and reliable and showed good construct and criterion validity with respect to the 36-Item Short Form Survey.

For the analyses, self-rated health was coded as ‘1’ if the response was poor or fair and coded as ‘0’ if the response was excellent, very good, or good. Poor physical health, poor mental health, and activity limitations were measured by the number of days. This number was then allocated to two response categories of 14 or more days (coded as 1), and less than 14 days (coded as 0).

#### 2.3.2. Bereavement

In its 2019 field survey, the Georgia BRFSS added a new module containing three items on the topic of bereavement. Participants were asked (1) ‘Have you experienced the death of a family member or close friend in the years 2018 or 2019?’. For those with a yes response, follow-up questions were asked ‘How many losses did you experience during that time?’ and ‘For each loss, please tell me if he or she was a spouse/partner, mother, father, brother, sister, child, other family members, friend, or neighbor, other, not sure, or refused.’ Bereavement was coded as ‘1’ (yes) and coded as ‘0’ (no). Number of losses reported was coded as follows: no losses were coded ‘0’, one loss, two losses, and three or more losses were coded as 1, 2, and 3, respectively. Relationship to the decedent was coded as, no losses were coded as 0, family member only (1), friend or neighbor (2), and others (3).

#### 2.3.3. Covariates

Covariates in the analyses were structured as categories—age groups (18–24 years, 25–34 years, 35–44 years, 45–54 years, 55–64 years, 65+ years), gender (male or female), self-reported race (black or African American only, white only, or all other), metropolitan area (metropolitan statistical county or non- metropolitan statistical county), educational attainment (did not graduate, high school, graduated, high school, attended college or technical school, or graduated, college or technical school), employment status (employed, unemployed, retired, unable to work, or homemaker or student), income group (less than $15,000, $15,000 to less than $25,000, $25,000 to less than $35,000, $35,000 to $50,000, or $ 50,000 or more), current smoking (yes or no), heavy drinking (yes or no), physical inactivity (yes or no), and obesity (non-obese, overweight, or obese).

### 2.4. Multiple Imputation of Missing Data

In Georgia, the median non-response rate for the BRFSS was 7.20% (ranging from 0 to 48.29%) for the variables used in this study. This is shown in Table A1. For these missing values, we adopted the missing at random assumption. Multiple imputation allows researchers to use more available data, thus reducing biases when observations with missing data are deleted [20]. Multiple imputation has three elemental phases: imputation, analysis, and pooling. The imputation phase was to create 50 copies of the dataset in this study, with the missing values replaced by imputed values using Multiple Imputation by Chained Equations (MICE). The MICE is a practical approach to impute missing data in multiple variables based on a set of univariate imputation models [21]. The variables listed in Table A1 were used in the imputation models. In the analysis phase, each of the 50 complete datasets was analyzed using a desired statistical method. The results obtained from 50 completed datasets were combined into a single multiple-imputation result in the pooling phase.

### 2.5. Data Analysis

A descriptive analysis of bereavement includes various individual characteristics and HRQoL domains. The purpose of the descriptive analysis is to help identify potentially relevant variables affecting HRQoL and prevalence of bereavement. Pearson’s chi-square test was performed for univariable analysis.

Since the dependent variables were binary response variables (self-rated poor health status, poor physical health, poor mental health, and activity limitations), logistic regression models were performed to analyze the association of bereavement with HRQoL based on imputed data. The independent variable was bereavement, number of losses, or relationship to decedent, respectively. The final model was adjusted for gender, age, race/ethnicity, and several social determinants of health—metropolitan area, educational attainment, employment status, income group, current smoking, heavy drinking, physical inactivity, and obesity. The results are presented as odds ratios (ORs) along with 95% confidence intervals (CIs).

To examine the role of health behaviors on the relationship between bereavement and HRQoL, the current study employed the commonly known Baron and Kenny approach [22], which is adjusted to assess mediation based on imputed data. This adjustment procedure uses structural equation modeling to study a mediation path that allows for many extensions and summarized steps for testing mediation via structural equation modeling.

The conceptual model underlying this analyses is derived from a ‘Shock’ frame of reference and its association with health spillovers [23,24]. Therefore, we hypothesize several channels (health behaviors) through which bereavement may impact HRQoL, drawing on theoretical evidence from the epidemiological literature. We graphically represent this conceptual framework in Figure 1. Our analyses considered health behaviors (smoking, heavy drinking, obesity, and physical inactivity) as the main mediating variables for the following reasons: (1) bereavement is associated with negative health behaviors [24,25] and (2) health behaviors were significantly associated with HRQoL among U.S. adults [26,27]. Health behaviors, bereavement, and HRQoL were binary variables and used generalized structural equation modeling rather than structural equation modeling for mediation analysis. Inference (standard errors and *p*-values) about indirect and total effects was performed using a nonlinear combination [28] The results are presented as coefficients (Coef.) along with 95% confidence intervals (CIs). All statistical analyses were conducted using Stata Version 17 (StataCorp, College Station, TX, USA).

## 3. Results

The sample size of the 2019 Georgia BRFSS was 7354 with 45.5% of respondents reporting one or more losses within 2018 or 2019. The total sample reflects the demographic composition of Georgia adults with 43.91% males and 34.08% aged 65 years or older. In the state, 71.2% lived in metropolitan counties. Approximately 88% of the respondents completed at least a high school education. Nearly 43% of the respondents reported an income of $50,000 or more.

Table 1 compares the subgroup of bereaved people with non-bereaved people according to demographics, social determinants of health, and health behaviors. Compared to non-bereaved people, bereaved people were (1) more likely to be aged 31 years or older, (2) more likely to be a woman, (3) Black or African American, (4) more likely to live in a non-metropolitan statistical county, and (5) less likely to have education beyond high school. The selected negative health behaviors were more common among bereaved people. These behaviors include current smoking, heavy drinking, and obesity.

Table 2 compares the unadjusted prevalence of reporting of the four HQRoL domains across the two categories (row headers). Bereavement is shown across two column headers—bereaved and non-bereaved people. This organization gives the reader a two-by-two table perspective for each domain. To interpret these rates, it is important to recall that the BRFSS survey item is worded to capture the immediate 30-day period prior to the interview. The table begins with two categories of self-rated health row domains—combined Poor and Fair, combined Excellent, Very Good and Good. The risk difference is calculated to compare the combined bereaved poor/fair group with the non-bereaved poor/fair. In the following rows, domains are shown as yes versus no responses. In each domain, bereavement is associated with an increased probability of a negative response. The risk difference associated with bereavement ranges from 2.8% (activity limitation) to 4.7% (mental health). These risk differences seem modest. However, when applied to the weighted population of BRFSS, the increased numbers of affected people can be estimated. To illustrate the population prevalence of bereavement and poor or fair self-rated health, consider the estimated 3.7 million adults bereaved in the 2019 survey. According to Table 2, 24.9% of this group report poor or fair self-rated health which translates into 921,300 people in the period prior to survey.

Table 3 shows the adjusted odds ratio for bereavement and HRQoL domains derived from adjusted logistic regression models. The dependent variables are self-rated poor health status, poor physical health, poor mental health, and activity limitations, while the independent variable was bereavement (Model 1), number of losses (Model 2), or relationship to decedent (Model 3), respectively. Adjustments include gender, age, race, social determinants of health, and health behaviors. In the 30 days prior to the survey, bereaved people reported significantly greater rates of poor mental health (41%, Model 1). Model 2 shows that rate of poor mental health significantly increased as the number of deaths increased—25% for one, 43% for two, and 67% for three or more. When there are three or more deaths reported, the rates of poor HRQoL items are increased—poor self-rated health (42%), poor physical health (38%), poor mental health (67%) and poor activity Limitations (41%). Model 3 measures the odds of poor HRQoL domains for categories of relationship. The risk for poor mental health is elevated with the death of a family member (28%). It is also significantly elevated with the death of a category labeled ‘Other’. Other is not specified. It could include professional caregivers or other people in the community such as work or school mates. The deaths of people in the Other category are associated with poor reports on three domains of HRQoL—24% self-rated health, 60% mental health, and 84% activity limitation.

Table 4 presents simple mediation models for bereavement and smoking (Model 1). Model 2 examines bereavement and obesity. The prevalence of these negative health behaviors are significantly elevated among the bereaved—Smoking (12.6% versus 9.7%), heavy drinking (6.1% versus 4.9%) and obesity (36.0% versus 31.7%). For the sake of clarity heavy drinking and physical activity are omitted from the table. The organization of the table emphasizes the effect of bereavement on each individual HRQoL domain as a ratio of indirect to total effect (RIT%). In Model 1, the ratios of indirect effect to total bereavement on HRQoL domains were 56.8% (poor self-rated health), 48.9% (poor physical health), 48.5% (poor mental health) and 54.4% (activity limitation). In contrast, in the obesity model (Model 2), the RIT values ranged from 40.7% (poor self-rated health), 28.1% (poor physical health), 14.3% (poor mental health) and 29.0% (activity limitation). Bereavement appears to play a significant but indirect role in mediating the association between negative health behaviors like smoking or obesity. By addressing bereavement within the context of smoking cessation or obesity reduction, a positive effect on population rates of smoking and obesity may also be achieved.

## 4. Discussion

Although bereavement—the death of a close friend or family member—is a feature of human existence, there is little scholarship on mass bereavement and its association with population-level health. Using an Economic Shock Model with Health Spillover [23,24] has helped shape the analyses presented in this paper. There are other examples in the literature. Resilience after disasters such as earthquakes [12,29,30] is commonly studied in this conceptual framework. By calculating societal risk, the Dutch government uses a conceptually similar estimate to revise safety policies [31].

Other aspects of the work require further investigation. By focusing on population-level bereavement, the present analyses cannot distinguish between typical grieving and clinical categories of bereavement i.e., prolonged grief disorder (PGD) and complicated grief (CG). From small samples, there is evidence that only seven to 10% of the population will progress to PGD and/or CG [32]. Other studies suggest that 66.4% of people with complicated grief recovered at the one-year mark and 25% of those with initially elevated grief recovered in the period of 6–12 months post-bereavement [33,34,35,36]. Comparison of the current work cannot be completed because these studies of PGD and CG are limited to a specific cause or a specific setting of clinical care. They do not contain a comparative sample of people without bereavement within a similar time frame.

This report advances science by measuring baseline bereavement as a component of an annual surveillance survey. Bereavement is highly prevalent among Georgia’s adults—3.4 million people in a population of 8 million [18]. Unlike prior studies of HRQoL and bereavement, BRFSS is not biased by preselection factors. It includes a representative sample of adults aged 18 years and older in a defined geographic area. Unlike other reports, it is not limited by the relationship between the decedent and the bereaved. Finally, it is not limited to a selected cause of death. BRFSS is designed for the development of population estimates by public health agencies. The unweighted denominator used in this analysis to develop estimates is an acceptable substitute when exploring mechanisms like bereavement and poor HRQoL. These analyses also calculate standard error which is used for calculating the confidence interval. The standard errors reported here are within the limits for reporting set by the Centers for Disease Control [37] The estimated increase in risk for poor HRQoL of three to five percent seems modest at first glance. However, when this proportion is applied to the size of the population exposed (3.4 million), the range in numbers of people needing clinical support (100,000 to 170,000) in a single state rapidly increases. Consider the costs and resources needed by public health authorities to respond to a grief associated outbreak. The elevated risk seems less modest under those circumstances.

In addition to SRHp, poor mental health is highly prevalent. These results suggest an axis for public action that includes targeting unhealthy health behaviors with a focus on bereavement care. Further studies are required because poor HRQoL is known to be highly population specific. When comparing 2019 Georgia (GA) with the 2019 Puerto Rico (PR), PR had higher rates of poor self-rated health (27%) versus 24.9% for GA, lower rates of poor physical health (12%) versus 18% for GA, lower rates of poor mental health for PR (11%) versus 17.5% for GA, and high rates of activity limitation for PR (19%) versus 14.8% for GA. This suggests that further evaluations of bereavement and HRQoL should be undertaken within the geographic context of a population. Bereavement was not measured in the 2019 Puerto Rico BRFSS.

This study uses a cross-sectional design and incorporates a new model module for monitoring bereavement in the 2019 Georgia BRFSS. To identify resilience factors to support specific public health actions requires repeated assessment of bereavement in subsequent surveillance surveys. For example, the mediation analyses suggests that bereavement plays a role in the prevalence of smoking and obesity. These two behaviors are targets for public health action. Recognition of bereavement and its contribution to poor HRQoL may improve the effectiveness of smoking cessation and obesity reduction. Assessment of bereavement within the 24 months prior to the surveillance survey is new. Placing bereavement within a limited time frame may reduce bias. The BRFSS, as with all annual surveillance surveys, conducts interviews via a telephone-based survey. This strategy does not appear to impact the likelihood of missing data and the size of standard errors [18]. These trends do not appear to be different across demographic groups within the sample—i.e., there is little or no bias due to refusal to respond to the item [18]. In the logistic and mediation models, there is adjustment for a wide variety of covariates. Adjustment is a possible source of confounding that will require further examination. Large health-related surveys, such as the BRFSS, provide many items capturing health-related risk behaviors and outcomes. Despite low rates of missing responses, multiple imputation recovers a fully observed sample size. More importantly, multiple imputation restores the natural variability of the missing values. Information recovery and restoring variability may reduce bias or increase precision, which results in a valid statistical inference from the imputation [36,37]. Finally, confirmation of the occurrence of a known death related to individual reporting is needed. The connection between a decedent and the bereaved has been tested using the National Mortality Follow Back Survey (NMFS) (https://www.cdc.gov/nchs/nvss/nmfs.htm, accessed on 8 December 2024). The last NMFS was conducted in 1993. Perhaps a return to this surveillance method is required.

Despite these limitations, these data show that a bereaved population is more likely to report poor HRQoL. By including bereavement in a surveillance survey, we provide a new perspective on a factor influencing mental health and the co-occurrence of detrimental health behaviors. This observation requires further examination. Studies of smaller samples clearly show that bereaved people report physical health complaints, higher rates of health care utilization, and accelerated mortality [13,16,18].

## 5. Conclusions

This study is a formal assessment of the effects of bereavement on HRQoL. It represents an improvement over prior studies. The sample includes a representative group of adults aged 18 years and older. All participants were asked to report any bereavement in the prior 24 months independent of its cause. When ascertained in this way, bereavement is significantly associated with poor HRQoL. The conceptual model for this study—Shock with health spillover—illustrates a path to poor HRQoL. A bereavement care policy designed to reduce health spillover may reduce the population-wide prevalence of poor mental health and protect population vitality.

## Figures and Tables

**Figure 1 behavsci-14-01213-f001:**
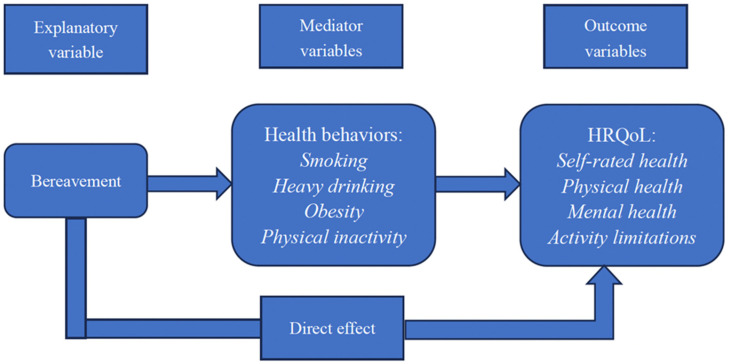
Conceptual Model. This model is derived from the Economic Shock Model and its association with health spillovers [23,24].

**Table 1 behavsci-14-01213-t001:** Differences between bereaved people and non-bereaved people according to demographics, social determinants, and health behaviors, 2019 BRFSS, Georgia.

	Bereavement	Non-Bereavement
	No. (%)	No. (%)
Observations	3347 (45.5)	4007 (54.5)
Demographics		
Age group		
18–24 years	198 (5.91)	**326 (8.14)**
25–34 years	343 (10.26)	**482 (12.02)**
35–44 years	456 (13.61)	494 (12.34)
45–54 years	**539 (16.11)**	587 (14.64)
55–64 years	**691 (20.65)**	732 (18.27)
65+ years	1120 (33.46)	1386 (34.59)
Gender		
Males	1399 (41.80)	**1830 (45.67)**
Females	**1948 (58.20)**	2177 (54.33)
Race		
Black or African American only	**915 (27.34)**	749 (18.70)
White only	2070 (61.84)	**2635 (65.75)**
All other	362 (10.82)	**623 (15.55)**
Socioeconomic Determinants		
Metropolitan area		
Metropolitan statistical county	2343 (69.99)	**2893 (72.21)**
Non-metropolitan statistical county	**1004 (30.01)**	1114 (27.79)
Educational attainment		
Graduated, College or Technical School	1082 (32.32)	**1429 (35.67)**
Attended College or Technical School	**953 (28.48)**	1037 (25.89)
Graduated, High School	**920 (27.49)**	1039 (25.92)
Did not graduate, High School	392 (11.71)	502 (12.52)
Employment status		
Employed	1545 (46.16)	1889 (47.14)
Unemployed	173 (5.16)	163 (4.07)
Retired	982 (29.33)	1205 (30.07)
Unable to work	382 (11.42)	375 (9.37)
Homemaker or student	265 (7.93)	375 (9.35)
Income group		
Less than $15,000	437 (13.07)	521 (12.99)
$15,000 to less than $25,000	714 (21.34)	782 (19.51)
$25,000 to less than $35,000	369 (11.02)	451 (11.26)
$35,000 to $50,000	434 (12.97)	509 (12.70)
$ 50,000 or more	1393 (41.60)	**1754 (43.54)**
Health Behaviors		
Current smoking		
Yes	**422 (12.62)**	391 (9.75)
No	2925 (87.38)	3616 (90.25)
Heavy drinking		
Yes	**204 (6.09)**	198 (4.93)
No	3143 (93.91)	3809 (95.07)
Physical inactivity		
Yes	1044 (31.19)	1251 (31.21)
No	2303 (68.81)	2756 (68.79)
Obesity		
Non-obese	1021 (30.51)	**1327 (33.13)**
Overweight	1121 (33.50)	**1408 (35.13)**
Obese	**1205 (35.99)**	1272 (31.74)

Note: Pearson’s chi-square test, Bold statistically significant *p* < 0.05.

**Table 2 behavsci-14-01213-t002:** Differences between bereaved people and non-bereaved people according to HRQoL domains.

	Bereaved	Not-Bereaved	Risk Difference	95% CI	*p* Value
	No. (%)	No. (%)			
HRQoL Domain	3347	4007			
Self-rated health					
Poor or fair	835 (24.94)	865 (21.58)	3.36	1.40–5.33	<0.05
Excellent, very good, or good	2512 (75.06)	3142 (78.42)			
Physical Health not good, 14 + days					
Yes	614 (18.33)	595 (14.86)	3.47	1.74–5.21	<0.05
No	2733 (81.67)	3412 (85.14)			
Mental Health not good, 14 + days					
Yes	586 (17.51)	513 (12.80)	4.71	3.04–6.38	<0.05
No	2761 (82.49)	3494 (87.20)			
Activity limitations, 14 + days					
Yes	497 (14.85)	483 (12.06)	2.79	1.21–4.39	<0.05
No	2850 (85.15)	3524 (87.94)			

Note: Pearson’s chi-square test, Risk difference = (Percent_bereaved_ − Perecent_notbereaved_).

**Table 3 behavsci-14-01213-t003:** Logistic regression models, bereavement: number of losses, and relationship to decedent associated with HRQoL domains, unweighted data with multiple imputation.

	Self-Rated Poor Health Status Adjusted OR (95% CI)	Poor Physical Health Adjusted OR (95% CI)	Poor Mental Health Adjusted OR (95% CI)	Activity Limitations Adjusted OR (95% CI)
Model 1				
Bereavement				
No	Reference	Reference	Reference	Reference
Yes	1.13 (0.98–1.32)	**1.22** **(1.02–1.45)**	**1.41** **(1.19–1.68)**	1.21 (0.98–1.48)
Observations	7354	7354	7354	7354
Model 2				
Number of losses				
0	Reference	Reference	Reference	Reference
1	0.99 (0.82–1.20)	1.15 (0.92–1.42)	**1.25** **(1.01–1.55)**	0.99 (0.76–1.29)
2	1.11 (0.88–1.41)	1.18 (0.90–1.54)	**1.43** **(1.11–1.85)**	**1.39** **(1.03–1.87)**
≥3	**1.42** **(1.14–1.77)**	**1.38** **(1.07–1.78)**	**1.67** **(1.30–2.16)**	**1.41** **(1.06–1.87)**
Observations	7354	7354	7354	7354
Model 3				
Relation to decedent				
No losses	Reference	Reference	Reference	Reference
Family only	1.02 (0.85–1.23)	1.14 (0.93–1.39)	**1.28** **(1.03–1.58)**	1.18 (0.88–1.57)
Friend or neighbor	1.11 (0.82–1.50)	1.34 (0.97–1.85)	1.39 (0.98–1.96)	0.94 (0.62–1.42)
Others	**1.24** **(1.01–1.53)**	1.26 (0.99–1.59)	**1.60** **(1.28–2.00)**	**1.84** **(1.36–2.49)**
Observations	7354	7354	7354	7354

Note: Odds ratio is adjusted for gender, age, race, metropolitan area, educational attainment, employment status, income group, current smoking, heavy drinking, physical inactivity, and obesity. Bold indicates statistical significance, *p* < 0.05.

**Table 4 behavsci-14-01213-t004:** Mediation Models for Bereavement and HRQoL domains.

	Poor Self-Rated Health Coef._unadj_ (95% CI)	Poor Physical Health Coef._unadj_ (95% CI)	Poor Mental Health Coef._unadj_ (95% CI)	Activity Limitations Coef._unadj_ (95% CI)
Model 1				
Bereavement → smoking	0.29 (0.12–0.46)	0.29 (0.12–0.46)	0.29 (0.12–0.46)	0.29 (0.12–0.46)
Smoking → HRQoL	0.72 (0.55–0.88)	0.77 (0.59–0.95)	1.10 (0.93–1.28)	0.85 (0.66–1.05)
Bereavement → HRQoL	0.17 (0.04–0.30)	0.23 (0.08–0.38)	0.34 (0.17–0.50)	0.21 (0.03–0.39)
Indirect effect	0.21 (0.08–0.34)	0.22 (0.08–0.36)	0.32 (0.12–0.52)	0.25 (0.09–0.41)
Total effect	0.37 (0.20–0.55)	0.45 (0.25–0.65)	0.66 (0.41–0.90)	0.46 (0.22–0.70)
RIT (%)	56.76	48.89	48.48	54.35
Model 2				
Bereavement → obesity	0.19 (0.07–0.31)	0.19 (0.07–0.31)	0.19 (0.07–0.31)	0.19 (0.07–0.31)
Obesity → HRQoL	0.57 (0.45–0.69)	0.47 (0.33–0.60)	0.34 (0.20–0.48)	0.45 (0.30–0.61)
Bereavement → HRQoL	0.17 (0.04–0.29)	0.23 (0.08–0.38)	0.36 (0.20–0.51)	0.22 (0.04–0.40)
Indirect effect	0.11 (0.04–0.18)	0.09 (0.03–0.15)	0.06 (0.02–0.11)	0.09 (0.02–0.15)
Total effect	0.27 (0.13–0.42)	0.32 (0.16–0.48)	0.42 (0.25–0.59)	0.31 (0.12–0.50)
RIT (%)	40.74	28.13	14.29	29.03

Note: RIT: ratio of the indirect effect to the total effect. Coef._unadj_ = Unadjusted coefficients.

## Data Availability

The data are available in a public repository. The Georgia Department of Public Health makes the data available https://gadph.justfoia.com/publicportal/home/track (accessed on 1 January 2020). There is no accession number. Request Georgia 2020 Behavioral Risk Factor Surveillance Survey (BRFSS). The Georgia Bereavement Module is available through the GA Department of Public Health.

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
