# Peer review of "Health-Related Quality of Life and Bereavement in the 2019 Georgia Behavioral Risk Factor Surveillance System"

_behavsci, 2024, doi:10.3390/bs14121213_

Round 1
Reviewer 1 Report
Comments and Suggestions for Authors
The authors have done a creditworthy job in demonstrating the health related correlates associated with bereavement experiences. Some of the associations noted could be related to social class differences between the bereaved and the non bereaved. An examination of the social class confounders with bereavement should be undertaken to better clarify how social class and bereavement many contribute to health difficulties.
Author Response
Please see attached letter

Reviewer 2 Report
Comments and Suggestions for Authors
This is an interesting use of the Behavioral Risk Factor Surveillance Survey in one US state to examine the health related correlates of recent bereavement .
Specific concerns include the following:
-Unclear why the title includes “prepandemic” as 2019 was clearly before the pandemic and there is not a comparison to mid or post-pandemic
-Abstract: “we define a group within representative population exposed to the death of a friend or relative in 2018 or 2019”- unclear what it means to define a group within representative population.
-Introduction: “When HRQoL is applied to individuals, it is a self-assessment.”- Unclear what this means.
-The term “effects of mass bereavement” is used but this terminology does not seem appropriate given it is looking at individual bereavement responses in a population.
-Methods: “cross-sectional and telephone interview” should eliminate the and
-because only 5,206 people responded to the bereavement questions, I would use that as the n for the study and in the abstract.
-Were non-respondents to the bereavement questions imputed when missing? If so, I would directly describe that because it is the independent variable.
- gender (males or females)- if you are only using male/female, consider using the term sex instead of gender
Conclusion
-It seems that bereavement is more likely in an older population. Older populations are more likely to be obese and smoke and have lower quality of life. It would be helpful to discuss this association.
Comments on the Quality of English Language-No concerns. Minor editing needed in some places (see comments)
Author Response
Please see attached letter

Reviewer 3 Report
Comments and Suggestions for Authors
The authors report a useful comparison of self-assessed quality of life as predicted by bereavement status (loss of a significant other in the last 24 months, yes or no), and the potential mediation of this effect by relevant health behaviors (especially smoking and obesity). By recruiting a representative population based sample rather than a convenience sample, the authors provide an unbiased assessment of these effects, contributing to our understanding of bereavement as a public health risk factor.
Two minor revisions would improve this worthy manuscript. First, it would be helpful to note more explicitly the seemingly modest increment in risk associated with bereavement, per se, as distinct from prolonged grief; differences attributable to loss of a relevant other vary from 2-5%. How relevant this magnitude of effect is to public health initiatives deserves some discussion.
Second, the text of the paper seems to be missing the occasional preposition or other word, and also introduced the occasional misplaced comma, making some sentences a bit of a challenge to decipher. A simple revision by a good editor who is not one of the authors could polish this minor blemish.
Author Response
Please see attached letter
